# Merging High-Resolution Satellite Surface Radiation Data with Meteorological Sunshine Duration Observations over China from 1983 to 2017

**Fei Feng** [1,*,†] **and Kaicun Wang** [2,†]

1 Research Center for Urban Forestry, College of Forestry, Beijing Forestry University, Beijing 100083, China
2 State Key Laboratory of Earth Surface Processes and Resource Ecology, College of Global Change and Earth System Science, Beijing Normal University, Beijing 100875, China; kcwang@bnu.edu.cn
\* Correspondence: forgetbear@bjfu.edu.cn or rememberbear@mail.bnu.edu.cn
† These authors contributed equally to this work.

**Abstract:** Surface solar radiation ($R_s$) is essential to climate studies. Thanks to long-term records from the Advanced Very High-Resolution Radiometers (AVHRR), the recent release of International Satellite Cloud Climatology Project (ISCCP) HXG cloud products provide a promising opportunity for building long-term $R_s$ data with high resolutions (3 h and 10 km). In this study, we compare three satellite $R_s$ products based on AVHRR cloud products over China from 1983 to 2017 with direct observations of $R_s$ and sunshine duration (SunDu)-derived $R_s$. The results show that SunDu-derived $R_s$ have higher accuracy than the direct observed $R_s$ at time scales of a month or longer by comparing with the satellite $R_s$ products. SunDu-derived $R_s$ is available from the 1960s at more than 2000 stations over China, which provides reliable decadal estimations of $R_s$. However, the three AVHRR-based satellite $R_s$ products have significant biases in quantifying the trend of $R_s$ from 1983 to 2016 ($-4.28$ W/m$^2$/decade to 2.56 W/m$^2$/decade) due to inhomogeneity in satellite cloud products and the lack of information on atmospheric aerosol optical depth. To adjust the inhomogeneity of the satellite $R_s$ products, we propose a geographically weighted regression fusion method (HGWR) to merge ISCCP-HXG $R_s$ with SunDu-derived $R_s$. The merged $R_s$ product over China from 1983 to 2017 with a spatial resolution of 10 km produces nearly the same trend as that of the SunDu-derived $R_s$. This study makes a first attempt to adjust the inhomogeneity of satellite $R_s$ products and provides the merged high-resolution $R_s$ product from 1983 to 2017 over China, which can be downloaded freely.

**Keywords:** surface solar radiation; sunshine duration; AVHRR; data fusion

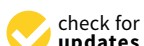



## 1. Introduction

Long-term changes in surface solar radiation ($R_s$, spectral range of 0.3–3.0 μm) with a decrease (global dimming) from the 1950s to the late 1980s and a subsequent increase (global brightening) have been observed worldwide [1–5]. Global dimming and brightening inevitably lead to energy balance changes and have substantial impacts on land surface climate, hydrologic cycle, carbon cycle and human activity [6–9]. However, high-quality $R_s$ observations are very sparsely distributed.

Direct observations can provide accurate $R_s$ records. Careful calibration and instrument maintenance are needed to guarantee the accuracy of observations. Before 1990, the USSR pyranometer imitations had different degradation rates in the thermopile, resulting in an important sensitivity drift [10–12]. To overcome radiometer aging, China replaced its instruments from 1990 to 1993. However, the new solar trackers frequently failed and introduced a high missing data rate for the direct radiation component of $R_s$ [13,14]. After 1993, although the instruments were substantially improved, the Chinese-developed pyranometers still had a high thermal offset with directional response errors, and the stability of these instruments was also worse than that of the World Meteorological Organization (WMO)-recommended first-class pyranometers [13,15,16].

Since the 1960s, the possibility of estimating $R_s$ from satellites has been examined in many studies [17,18]. To date, satellite measurement records have grown to cover more than 40 years. In 1983, the International Satellite Cloud Climatology Project (ISCCP) began its operational phase [19]. Many long-term satellite-based climatologies of cloud cover and cloud radiative property data have been released, such as the ISCCP [20], the PATHFINDER Atmospheres-Extended (PATMOS-x; [21]), Maryland version of the Global Energy and Water Cycle Experiment (GEWEX; [22]), EUMETSAT Climate Monitoring Satellite Application Facility (CMSAF) cloud and radiation data record (CLARA-A2; [23]) and the Climate Change Initiative Cloud project (Cloud_cci; [24]) funded by the European Space Agency (ESA). Due to the capability of capturing the spatial distribution and dynamic evolution of clouds from satellite data [10,25], satellite-derived $R_s$ products have long been recognized as better methods to estimate regional and global $R_s$ than numerical modeling [26,27]. Improvement have also been shown in $R_s$ data from DISORT radiation transfer model of the China Meteorological Administration Land Data Assimilation System (CLDAS) by assimilating satellite $R_s$ retrievals from FY2E and ISCCP cloud data [28].

The trend of $R_s$ derived from satellites has been shown to be consistent with ground observations but differs in sign and magnitude [29]. Satellite $R_s$ products based on the early version of the ISCCP cloud dataset show quite different interannual variations in $R_s$ from 1984 to 2000, which are only approximately one half to one third of the direct $R_s$ surface observations over China [30]. The assessment of cloud fraction from ISCCP, PATMOS-x and CLARA-A1 for 1982–2009 over the United States indicates that the ISCCP tends to show less year-to-year variability, while PATMOS-x and CLARA-A1 tend to show higher variability [31]. Wang et al. (2018) [31] also noticed that when neglecting temporal aerosol variability in the satellite algorithms in CLARA-A1 and SARAH-E, the discrepancy between the satellite-estimated and ground-observed SSR trends slightly increased over China in 1999–2015 compared with 1993–2015.

Efforts have been made to obtain optimized $R_s$ data and produce reliable historical $R_s$ data records [4,32,33]. Various methods used to identify and adjust these artifacts in the $R_s$ records have been widely developed. These inhomogeneity test methods mostly require reference series constructed from highly correlated nearby stations [34]. For example, the standard normal homogeneity test (SNHT), Pettitt test, Buishand test and Von Neumann test have been applied to adjust the $R_s$ data in the Global Energy Balance Archive (GEBA) dataset due to the impacts of station relocation and instrument change [35]. Relative tests developed by (Brunetti et al. 2006) [36] were used to correct the Italian $R_s$ data before 1980 [34,37].

The surface sunshine duration (SunDu) is the sum of the time for which the ground surface is irradiated by a direct solar irradiance beam exceeding a certain threshold (usually 120 W/m$^2$) and is considered to be a reliable proxy of $R_s$ [38–42]. Since SunDu is widely recorded at many stations, Tang et al. (2013) [32] extended $R_s$ records to 716 China Meteorological Administration (CMA) routine weather stations from 96 radiation stations based on an artificial neutral network model. SunDu has been used as reference data to adjust the inhomogeneity of direct observations over China. With the improved Angstrom–Prescott model, Tang et al. (2013) [32] show that SunDu data can derive accurate estimates of hourly, daily and monthly $R_s$. Previous studies further demonstrate that SunDu can be used as a reliable proxy for direct $R_s$ observed data to reconstruct long-term $R_s$ [2,5,43–47]. By using global SunDu-derived $R_s$ records, He et al. (2018) [2] show that SunDu permitted a revisit of global dimming from the 1950s to the 1980s over China, Europe and the USA, with brightening from 1980 to 2009 in Europe and a declining trend of $R_s$ from 1994 to 2010 in China. Wang et al. (2015) [10] also found that the dimming trend from 1961 to 1990 and nearly constant zero trend after 1990 over China, as calculated from the SunDu-derived $R_s$, were consistent with independent estimates of AOD [48]; they also observed changes in the diurnal temperature range [49] and the observed pan evaporation [50].

The previously mentioned inhomogeneity adjustment methods mainly focus on $R_s$ data over observation sites, and the adjustment processes are generally "point to point"

corrections. Comparatively, inhomogeneities in the satellite $R_s$ gridded data have attracted little attention due to the lack of reliable $R_s$ reference data with large spatial coverage. As systematic $R_s$ biases in gridded data can be removed by simple bias correction methods, uncertainties in the long-term variations in the $R_s$ values of gridded data cannot be easily eliminated [51]. As long-term reliable aerosol data are rarely available, especially for data extended before 2000, improved gridded surface radiation data, such as satellite retrievals, will be more challenging.

Merging multisource data can be a practical way to improve $R_s$ data with accurate long-term variations [51–56]. The large number of SunDu observations (~2400 stations) with large spatial coverage (Figure 1) can provide substantial benefits to obtain better estimations of $R_s$ over China. On the other hand, based on the ISCCP-HXG cloud products and new ERA5 reanalysis data, Tang et al. (2019) [57] built global satellite $R_s$ products with a 10 km spatial resolution from 1983 to 2017. However, Tang et al. (2019) claimed that its application in quantifying long-term variability should be considered cautiously.

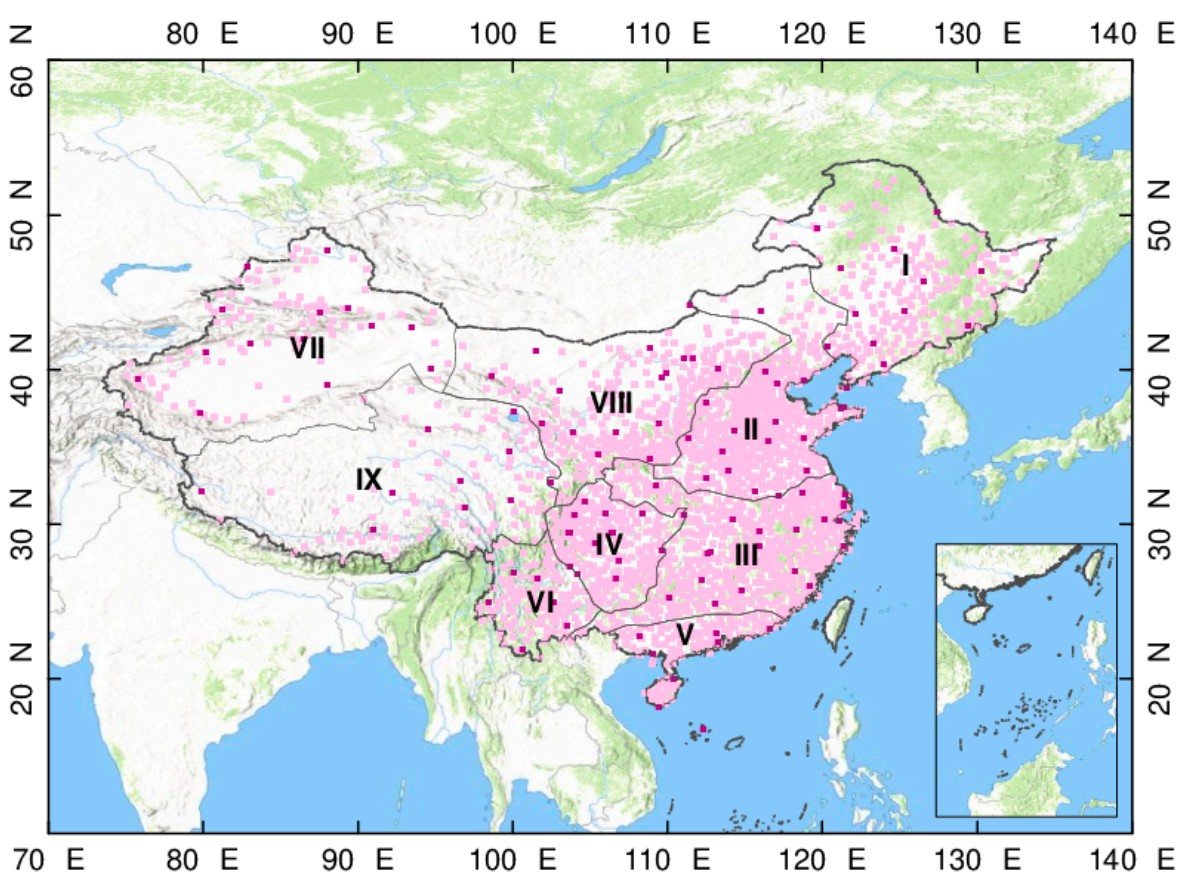

**Figure 1.** The ~2400 sunshine duration (SunDu) merged sites are shown as light pink points, and 121 independent validation sites are shown as dark pink points. The whole region is classified into nine subregions (I to IX) by the K-mean cluster method based on geographic locations and multiyear mean $R_s$ using 121 $R_s$ validation sites. The base hill shade map was produced by an elevation map of China using the global digital elevation model (DEM) derived from the Shuttle Radar Topography Mission 30 (SRTM30) dataset.

Our previous studies have shown reanalyses have substantial biases due to imperfect parameterization of cloud and aerosols [26,51,58]. Considering the advantages of SunDu-derived $R_s$ and reliable cloud distribution from satellite retrievals, we recently combined ground-based sunshine duration with satellite cloud and aerosol data to produce high resolution long-term $R_s$ [59]. Another approach is to directly merge current high resolution $R_s$ product with SunDu-derived $R_s$. Therefore, the purpose of this study is to develop a fusion algorithm based on a geographically weighted regression model to improve



current $R_s$ satellite gridded data by using surface sunshine duration-derived $R_s$ site data as references over China. First, we use direct $R_s$ observations and SunDu-derived $R_s$ to assess the performances of these long-term satellite $R_s$ products, including CMSAF CLARA-A2 (CMSAF), GEWEX-SRB (GEWEX) and ISCCP-HXG (HXG), over China, especially for the long-term $R_s$ trends. These comparisons demonstrate that SunDu-derived $R_s$ is a reliable data source and that satellite products have significant inhomogeneity in quantifying the long-term variability in $R_s$ over China. In terms of the longest time span (1983~2017) of ISCCP-HXG, we merge ISCCP-HXG with SunDu-derived $R_s$ to obtain the best estimate of $R_s$ over China.

## 2. Data and Methodology

### 2.1. Satellite Retrievals of Surface Solar Radiation

We collect three AVHRR based satellite monthly $R_s$ data including: (1) the Satellite Application Facility on Climate Monitoring cLoud, Albedo and surface RAdiation dataset from AVHRR data—Edition 2 (CMSAF CLARA-A2), (2) The Global Energy and Water Exchanges-surface radiation budget (GEWEX-SRB), (3) the level 2 International Satellite Cloud Climatology Project ISCCP H-series cloud product HXG (ISCCP-HXG) based $R_s$ product. In addition, Clouds and the Earth's Radiant Energy System energy balanced and filled product (CERES EBAF) surface product (Edition 4.1) are also used in this study. The brief summary of these satellite $R_s$ data are shown in Supplementary Material Table S1.

The retrieval algorithms of CLARA-A2 are the same as those of CLARA-A1, which is based on the mesoscale atmospheric global irradiance code (MAGIC) and related look-up-table algorithms driven with cloud data from CLARA-A2 and auxiliary data, including surface albedo, monthly averaged integrated water vapor and aerosol climatology derived from the Aerocom model median merged with AERONET in situ data [60,61].

GEWEX-SRB V3.0 estimates $R_s$ by using a modified radiation transfer model from [62]. The input cloud properties are derived from ISCCP DX data,; temperature and moisture profiles are derived from GMAO-GSFC [63], and ozone-column data are primarily from TOMS, TOVS and SMOBA analyses. Aerosol extinction and absorption by ozone and water vapor are calculated following (Pinker et al. 2003) [22].

The ISCCP-HXG-based $R_s$ product is retrieved based on the SUNFLUX scheme [64,65]. Input data are used to drive the model, including cloud mask, visible (VIS)-retrieved liquid cloud optical depth, VIS-retrieved ice cloud optical depth and cloud top temperature from ISCCP-HXG. Surface pressure, total column water vapor and total column ozone are from ERA5 reanalysis data. Aerosol and albedo data are from MOD08_D3 and MCD43A3. Missing values in the aerosol and albedo data were replaced with the corresponding values from the monthly mean climatological data.

CERES EBAF is the level 3b product based on the CERES radiometer-measured filtered shortwave radiances (0.3–5 μm, 0.3–200 μm and 8–12 μm). The cloud properties are collected from MODIS and geostationary imagers [66]. MODIS AOD C6.1 assimilated into an aerosol transport model (MATCH) are used as aerosol input data for the CERES EBAF product [67].

### 2.2. Ground-Based Estimations of Surface Solar Radiation

We used direct $R_s$ measurements from 121 radiation stations collected from 1983 to 2016 by the China Meteorological Data Service Center (CMDC, http://data/cma/cn/) of the CMA. Monthly $R_s$ values are calculated by averaging daily $R_s$ values when daily observed data are available for more than 15 days for each month at each radiation station (Figure 1).

Sunshine durations from 1983 to 2017 are obtained from approximately 2400 CMA weather stations. These sunshine durations are used to calculate monthly mean $R_s$ following the method of the revised Ångström-Prescott Equation (1) [10,11,68].

$$\frac{R_s}{R_c} = a_0 + a_1 \frac{n}{K} + a_2 \left(\frac{n}{K}\right)^2 \tag{1}$$

where $n$ is the measured sunshine duration and $K$ is the theoretical value of sunshine duration. $a_0$, $a_1$ and $a_2$ are determined according to (Wang 2014) [11]. $R_c$ is the daily total solar radiation under clear-sky conditions.

Sunshine duration-derived $R_s$ has been shown to be a good proxy of directly observed $R_s$ at time scales ranging from monthly to decadal and can reveal the impact of aerosols and clouds on $R_s$ over China [69–72]. The long-term coverage of the relatively widely distributed sunshine duration data permits the long-term trend analysis of $R_s$ with large spatial coverage [3,73]. In addition, SunDu-derived $R_s$ values are nearly free from the sensitivity drift problem, which hampers traditional $R_s$ observation data applications over China.

### 2.3. The Fusion Method

Geographically weighted regression (GWR) is used as a fusion method to merge the ISCCP-HXG-derived $R_s$ with ground-based SunDu-derived $R_s$ (HGWR). GWR is an extension of the traditional regression model by allowing the spatial modeling of nonstationary processes. The principle of GWR is provided as follows:

$$y_i = \delta(i) + \sum_k \delta_k(i) x_{ik} + \varepsilon_i \tag{2}$$

where $y_i$ is the value of the merged $R_s$ unit $i$; $i = 1, 2, \ldots, n$, $n$ denotes location $i$, $x_{ik}$ indicates the corresponding values of ISCCP-HXG-derived $R_s$, and $\varepsilon$ denotes the residuals. $\delta_{(i)}$ is the regression intercept. $\delta_{k(i)}$ is the regression coefficient calculated from the spatial weighting function $w_{(i)}$, which quantifies the proximities of location $i$ to its neighboring observation sites; $X$ is the variable matrix; and $b$ is the bias vector.

$$\delta_k(i) = \left( X^T w(i) X \right)^{-1} X^T w(i) b \tag{3}$$

Owing to the irregular distribution of observation sites and computer ability, the adaptive Gaussian function method is selected as a weighting function that varies in extent as a function of $R_s$ observation site density.

$$w_{ij} = \exp\left( -\left( d_{ij}/b \right)^2 \right) \tag{4}$$

where $w_{ij}$ is the weighting function for SunDu derived $R_s$ observation site $j$ that refers to location $i$; $d_{ij}$ denotes the Euclidian distance between $j$ and $i$; and $b$ is the size of the neighborhood and the maximum distance away from regression location $i$, known as "bandwidth", which is determined by the number of nearest neighbor points (NNPs). The NNPs are set to 30 following our parameterization experiment (Table S2, Feng and Wang, 2020 under review). The regression coefficients are then determined by Equations (3) and (4) using the ground-based SunDu $R_s$ as the dependent variable and ISCCP-HXG $R_s$ at corresponding observation site as the independent variable. The final merged $R_s$ can be obtain by the regression coefficient and ISCCP-HXG $R_s$ for the whole regions by Equation (1), which can be applied in ArcGIS or R. Previous studies have shown that GWR performs better than other regression methods, especially for analyzing spatial data [74–79].

### 2.4. Preprocessing and Metrics for Evaluation

All the 121 direct $R_s$ observation sites, where both direct observed $R_s$ data and SunDu-derived $R_s$ data are recorded, are used as validation sites, and the remaining 2261 sites are merged with HXG by GWR (Figure 1). We obtain monthly $R_s$ from three AVHRR-based $R_s$ data and CERES EBAF. HXG only provide 3 hourly $R_s$ estimates. We calculate daily mean $R_s$ and obtain monthly average estimates for the $R_s$ data. The CMSAF data contain a large amount of missing data before 2000. To avoid the impacts of missing data in CMSAF, we only use data with less than 20% of the data missing from the total number of validation sites.

To make inter-comparisons, we chose the years 2000 to 2007, a period common to all three AVHRR-based datasets and CERES EBAF. The determination coefficient ($R^2$), root mean square error (RMSE), bias error and mean absolute bias (MAB) are calculated for validation. When performing the trend analysis, all data from 1983 to 2016 are used if the product is available. $R_s$ validation results and long-term variation patterns are also presented for the nine subzones by the K-means clustering algorithm based on geographic locations and multiyear mean $R_s$ derived from all SunDu-derived $R_s$ sites. For comparisons of the spatial distribution of multiyear mean $R_s$ and long-term trends, the SunDu-derived $R_s$ are interpolated to $1° × 1°$ grids following (Du et al. 2017) [80]. For long-term $R_s$ variation evaluation, the linear trends in $R_s$ are calculated by the least squares method for different time periods, including 1983 to 2007, 2000 to 2007, 2000 to 2016 and 1983 to 2016.

## 3. Results

### 3.1. Site Validations

All these four AVHRR-based $R_s$ products showed high $R^2$ values ranging from 0.89 to 0.97 (Figure 2). Generally, HGWR show best performances followed by HXG, GEWEX and CMSAF. Specifically, CMSAF has the largest MAB and RMSE (18.96 W/m$^2$ and 24.08 W/m$^2$, respectively) followed by GEWEX and HXG, compared with direct observations. Similar results can also be seen when validating against the SunDu-derived $R_s$. When comparing with SunDu-derived $R_s$, GEWEX has the largest MAB and RMSE (17.46 W/m$^2$ and 22.17 W/m$^2$, respectively) followed by CMSAF and HXG (Table S3). The good performances of HXG might be attributed to the improvements of high resolution of cloud from latest ISCCP-H series product and auxiliary data from ERA5. By merging SunDu $R_s$ data, HGWR showed the lowest biases with MAB = 13.86 W/m$^2$ and RMSE = 19.16 W/m$^2$ compared with direct observations. Obvious improvements in HGWR are shown in comparison with SunDu $R_s$ reference data, with the MAB reduced by 33%. Similar results can also been seen in different seasonal (Figures S2–S5).

Generally, this is likely because satellite derived $R_s$ products are at grid of 1 degree, and both direct observation and SunDu are ground-based point observations. However, the time span of CERES EBAF is 2000 to present. As direct $R_s$ observation, SunDu derived $R_s$ and CERES are three independent measurements, the inter-comparison show that direct $R_s$ observation is neither consistent with CERES EBAF nor SunDu derived $R_s$ and the latter two show more consistent results, which again indicates that the SunDu derived $R_s$ can be used to produce reliable $R_s$ data in monthly time scale. The detailed validation results are summarized in Table S3.

The monthly validation results for each subzone (Figure 1) from 2000 to 2007 are shown in Figure S1. In subzone I, which is located on the North China Plain, GEWEX has the largest biases (MAB = 15.74 W/m$^2$) and the HGWR has the lowest (MAB = 7.20 W/m$^2$) compared with independent SunDu-derived $R_s$. These results suggest that without the SunDu derived $R_s$ constraint, these AVHRR based satellite $R_s$ retrievals still have uncertainties in simulating aerosols. Similar performances of GEWEX and HGWR are shown in the other eight regions, except that CMSAF has the largest MAB (16.06 W/m$^2$) in zone V. By merging the SunDu-derived $R_s$ site data, the MAB of HXG was reduced by 31% in zone I, 36% in zone II/III, 27% in zone IV, 31% in V, 32% in zone VI, 44% in zone VII, 33% in zone VIII and 22% in zone IX. The areas with significant improvement of HXG are mainly distributed in east part of China, which might be the improvement in simulating the impact of aerosols loading by merging SunDu derived $R_s$.

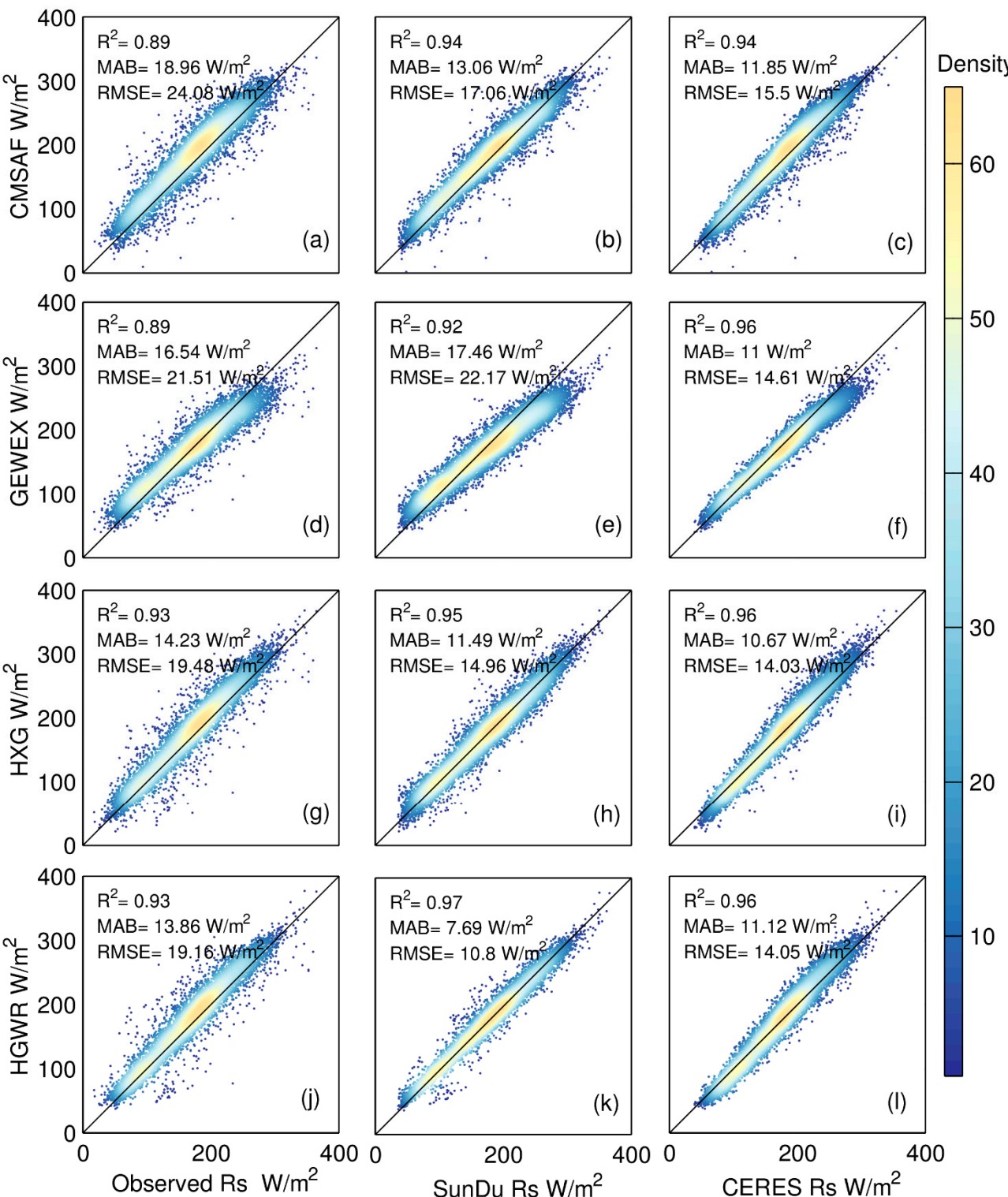

**Figure 2.** Comparison of monthly surface solar radiation ($R_s$) from the Climate Monitoring Satellite Application Facility (CMSAF)cloud and radiation data record (CLARA-A2) (CMSAF), Global Energy and Water Exchanges-surface radiation budget (GEWEX-SRB) (GEWEX), International Satellite Cloud Climatology Project ISCCP H-series cloud product HXG (ISCCP-HXG)-based $R_s$ data (HXG) and the merged product (HGWR) by using different validation data from 2000 to 2007. Subplots (**a**,**d**,**g**,**j**) represent validation results using direct observations, while (**b**,**e**,**h**,**k**) represent SunDu-derived $R_s$ data (SunDu $R_s$), and subplots (**c**,**f**,**i**,**l**) represent Clouds and the Earth's Radiant Energy System energy balanced and filled product (CERES EBAF) data (CERES).

### 3.2. Spatial Distribution

As shown in Figure 3, the multiyear means of $R_s$ from 2000 to 2007 derived from the SunDu sites are high in west China, ranging from 180 to 300 W/m$^2$, and low in eastern China, ranging from 120 to 180 W/m$^2$. CERES EBAF, CMASF, GEWEX, HXG and HGWR have similar spatial patterns of $R_s$. Slight differences are found in west China. HXG and HGWR show better performances than other $R_s$ products compared with the SunDu-derived $R_s$ reference data, which might be attributed to the high spatial resolution of cloud input data from ISCCP-HXG (Table S3). By merging the SunDu-derived $R_s$, HGWR showed an improved performance compared with HXG at independent SunDu sites, with R$^2$ values from 0.91 to 0.95 MAB reduced from 27.6% to 4.97 W/m$^2$ and RMSE reduced from 25.2% to 6.13 W/m$^2$ (Table S3).

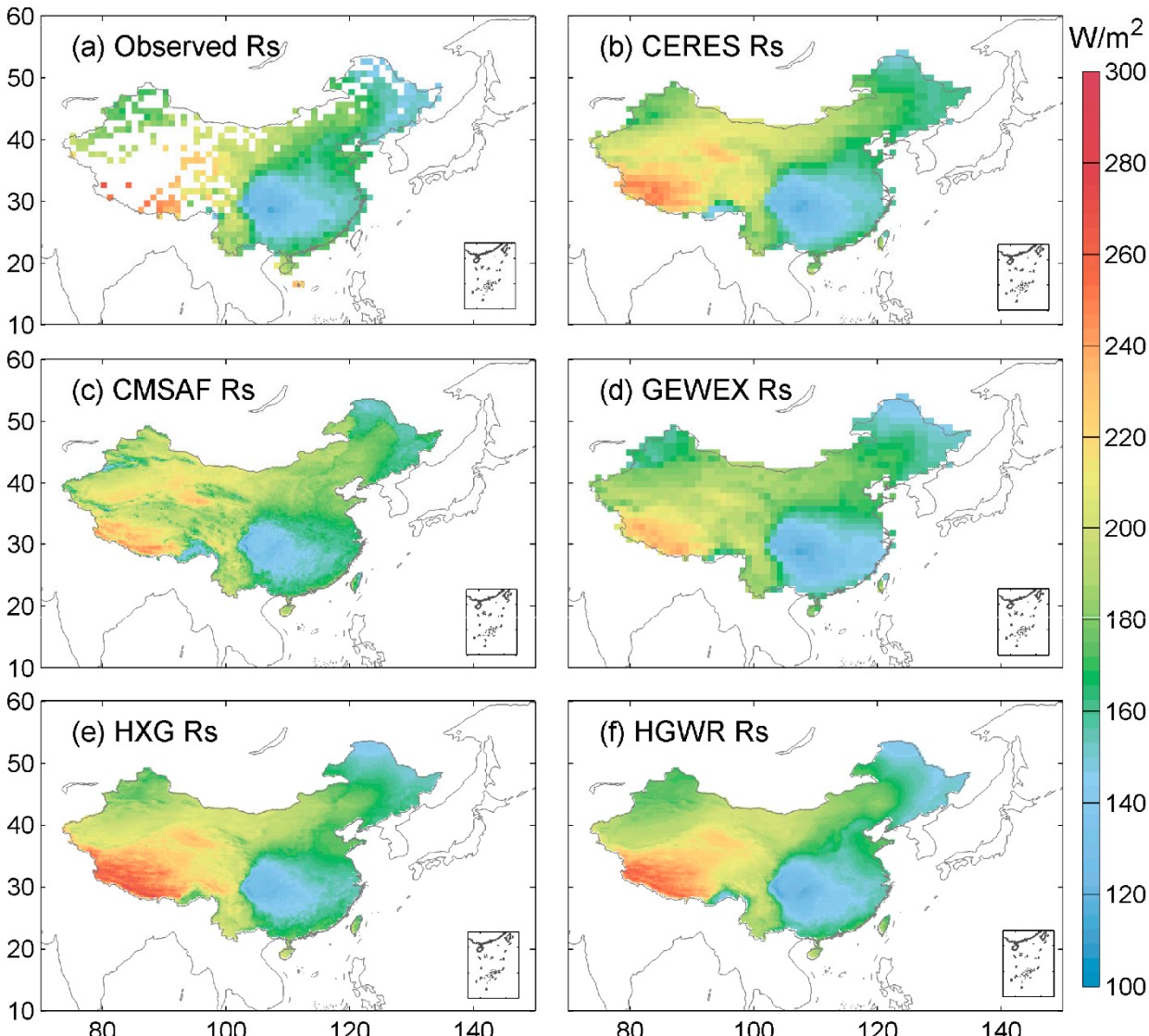

**Figure 3.** Spatial distribution of multiyear mean $R_s$ from 2000 to 2007. The first line (**a**,**b**) shows the observed multiyear mean $R_s$ from SunDu and CERES EBAF (CERES); the multiyear mean $R_s$ derived from the CMSAF CLARA-A2 (CMSAF) and GEWEX-SRB (GEWEX) are shown in the second line (**c**,**d**), respectively. The third line (**e**,**f**) shows the observed multiyear mean monthly $R_s$ from ISCCP-HXG (HXG) and ISCCP-HXG merged with SunDu-derived $R_s$ (HGWR).

The spatial distribution of trends in $R_s$ over China from 2000 to 2007 is shown in Figure 4. Based on the SunDu-derived $R_s$ data, $R_s$ shows an increasing trend in South China and a decreasing trend in most parts of the North China Plain. CERES EBAF and CMSAF

have consistent spatial distributions of $R_s$ trends over most parts of China compared with the SunDu-derived $R_s$. The results demonstrate that the SunDu-derived $R_s$ can produce reliable $R_s$ trends with large spatial coverage. GEWEX and HXG produce decreasing trends in most parts of China, which is inconsistent with the SunDu $R_s$ and CERES EBAF data. By merging the SunDu-derived $R_s$ site data, the inconsistent $R_s$ trends from HXG are corrected and compared with the CERES and CMSAF $R_s$ data. The difference spatial distribution of $R_s$ trend might be attributed to the input of cloud and aerosols data. Both CERES EBAF and CMSAF use CALIPSO-CALIOP cloud information to adjust the cloud input data. Both GEWEX and HXG use climatology of aerosol data [22]. In southern coastal regions of China, all $R_s$ products show decreasing trend, which might be the dominated impacts of clouds. However, in the southern inland regions of China, the $R_s$ trend from these $R_s$ products is quite different, which might be the different ability of simulating the aerosols loading in these regions. In the western part of China, largest differences in $R_s$ trend from these $R_s$ products show in the Qinghai-Tibet Plateau and Tianshan Mountains, which might be attributed to impacts of the complex terrain. Similar results also show in Daxinganling Mountain in Northeast China.

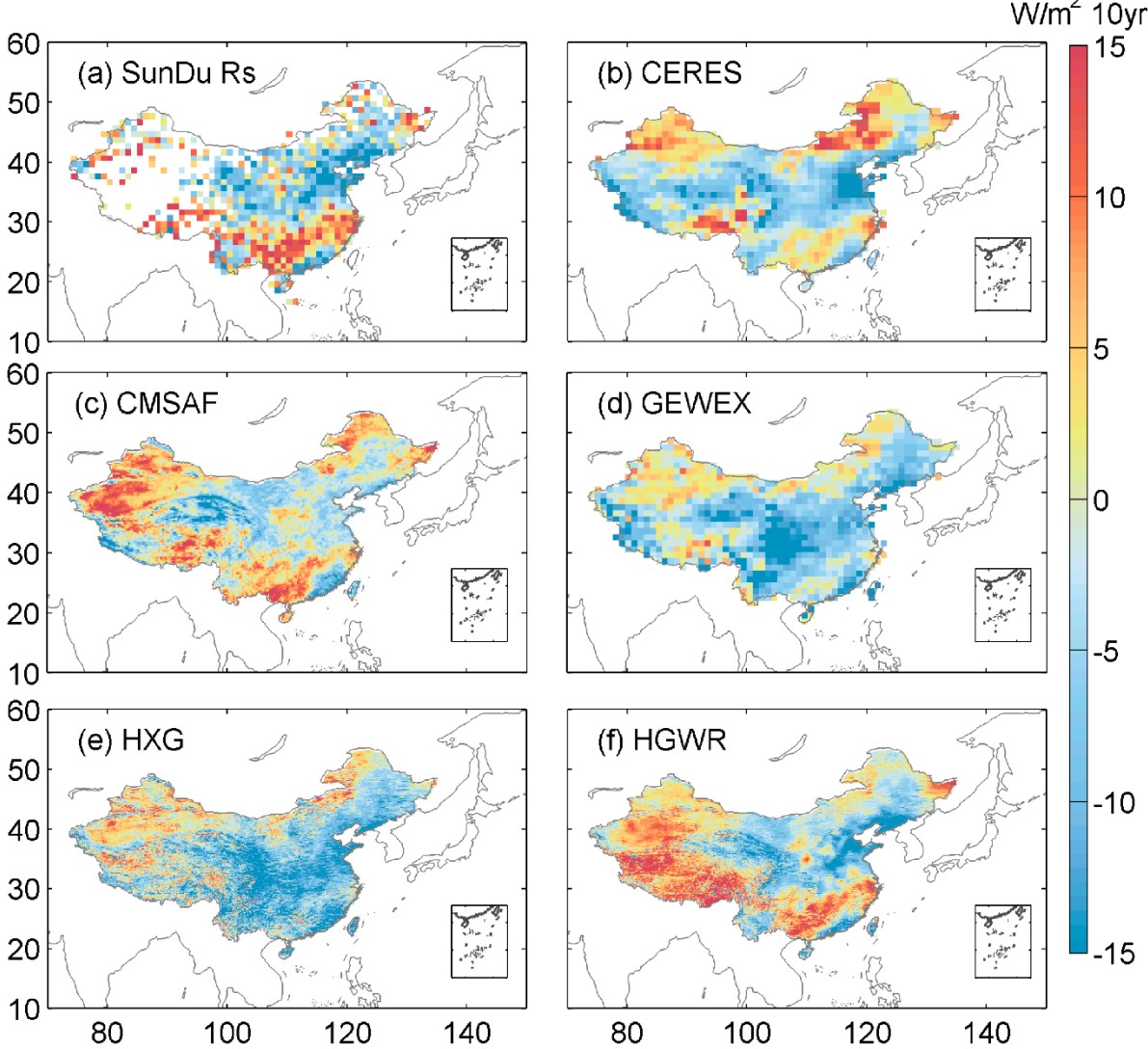

**Figure 4.** Spatial distributions of trends for $R_s$ from 2000 to 2007. The first line (**a**,**b**) shows the SunDu-derived $R_s$ and CERES EBAF $R_s$ (CERES); the $R_s$-derived CMSAF CLARA-A2 (CMSAF) and GEWEX-SRB (GEWEX) are shown in the second line (**c**,**d**). Subplots (**e**,**f**) represent ISCCP-HXG-derived $R_s$ (HXG) and ISCCP-HXG merged with SunDu-derived $R_s$ (HGWR).

To further assess the spatial distribution of the $R_s$ trend, we compared the spatial distribution of the $R_s$ trends over three time periods, including 1984 to 2000, 2000 to 2016 and 1984 to 2016 (Figure 5). For 1984 to 2000, SunDu-derived $R_s$ showed an increasing $R_s$ trend over central and northeastern China and a slight decreasing trend for the southern part of China. HXG overestimates the decreasing $R_s$ trend in southern China and underestimates the increasing $R_s$ trend over central and northeastern China. For 2000 to 2016, the SunDu-derived $R_s$ and HXG produced similar $R_s$ trends over China. For 1984 to 2016, the SunDu-derived $R_s$ showed a levelling off trend over most parts of China, with a slight increasing trend over Southwest China and a decreasing trend over the North China Plain. Comparatively, HXG produces an overestimated decreasing trend in most parts of China. By merging the SunDu-derived $R_s$ data, the inconsistent $R_s$ trends in these time periods are corrected in HGWR. The different spatial distribution of Rs trends between SunDu derived Rs and HXG might be that HXG use climatology aerosol data.

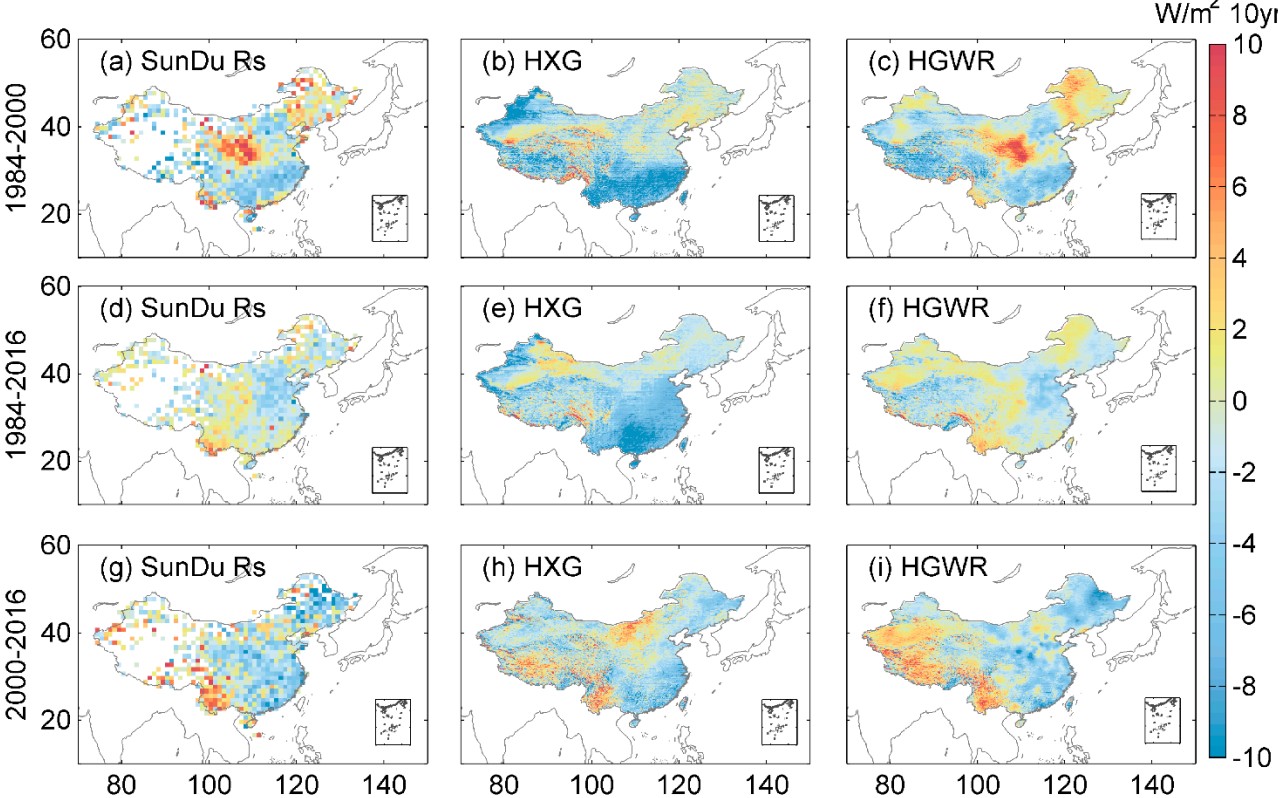

**Figure 5.** Spatial distributions of trends for $R_s$ from 1984 to 2000, 2000 to 2016 and 1984 to 2016. The first line (**a**,**b**,**c**) shows the $R_s$ trends from 1984 to 2000 based on SunDu-derived $R_s$ (SunDu Rs), ISCCP-HXG-derived $R_s$ (HXG) and ISCCP-HXG merged with SunDu-derived $R_s$ (HGWR). The second (**d**,**e**,**f**) and third lines (**g**,**h**,**i**) show the $R_s$ trends from 1984 to 2016 and 2000 to 2016, respectively.

### 3.3. Long-Term Variation

Figure 6 shows the national mean long-term variation in $R_s$ from observations and satellite retrievals. The directly observed $R_s$ showed a decreasing trend before 1990, an abrupt increase from 1990 to 1994, and an increasing trend after 2000. The SunDu-derived $R_s$ showed a decreasing trend before 1990 and a levelling off trend afterward. Comparatively, the SunDu-derived $R_s$, CERES EBAF and CMSAF produce consistent $R_s$ annual variations after 2000. Before 2000, large numbers of missing data eliminated the CMSAF to produce continuous $R_s$ variations over China. GEWEX has a nearly consistent $R_s$ variation compared with the SunDu-derived $R_s$, except it underestimates the $R_s$ variation from 2002 to 2004. HXG shows a large, inconsistent $R_s$ variation compared with the SunDu-derived $R_s$. By

merging the SunDu-derived $R_s$ data, the inconsistent $R_s$ variation in HXG is corrected in HGWR. Table 1 also shows that the HGWR produces the closest national mean trend $(-0.64 \text{ W/m}^2/\text{decade})$ from 1983 to 2016 compared with SunDu $(-0.57 \text{ W/m}^2/\text{decade})$.

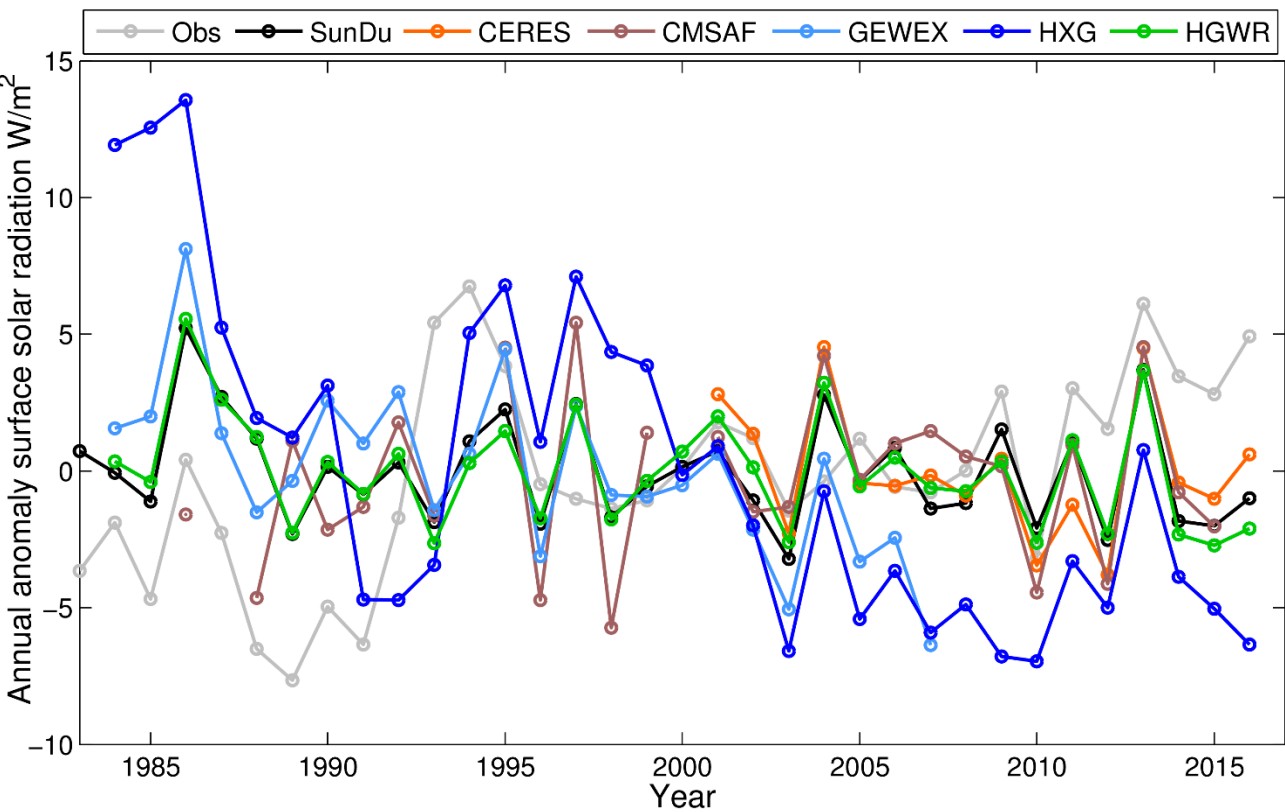

**Figure 6.** The national mean of the annual anomaly of the $R_s$ by using 121 observation sites from 1983 to 2016 derived from observations and satellite-derived $R_s$, including direct $R_s$ observations (Obs), SunDu derived $R_s$ (SunDu), CERES EBAF (CERES), GEWEX-SRB (GEWEX), ISCCP-HXG (HXG) and the merged $R_s$ data (HGWR). Direct observed $R_s$ data, SunDu-derived $R_s$ and CERES EBAF are shown as the light grey line, the black line and the orange line, respectively. The brown line, the light blue line and the blue line represent the $R_s$ variations derived from CMSAF CLARA-A2 (CMSAF), GEWEX-SRB (GEWEX) and ISCCP-HXG derived $R_s$ (HXG), respectively. The ISCCP-HXG derived $R_s$ merged with SunDu derived $R_s$ (HGWR) is shown as the green line.

**Table 1.** Statistical summary of national mean linear trends of monthly anomalous surface solar radiation ($R_s$) (10 years) derived from observations and satellite retrievals, including direct observations (Obs), SunDu-derived Rs (SunDu), CERES EBAF (CERES), GEWEX-SRB (GEWEX), CMSAF CLAR A2 (CMSAF), ISCCP-HXG (HXG) and merged Rs data (HGWR).

|  | 1983–2007 | 2000–2007 | 2000–2016 | 1983–2016 |
|---|---|---|---|---|
| Obs | −2.52 | −1.81 | 2.87 | −0.07 |
| SunDu | −0.77 | −0.76 | −0.26 | −0.57 |
| CERES |  | −3.25 | −1.25 |  |
| CMSAF | 5.46 | 1.36 | −1.16 | 2.56 |
| GEWEX | −2.88 | −6.96 | −6.96 | −2.88 |
| HXG | −5.28 | −8.40 | −2.12 | −4.28 |
| HGWR | −0.49 | −1.89 | −1.53 | −0.64 |

Based on the classified subregions shown in Figure 1, we further evaluated the regional mean long-term variation in $R_s$ from the observations and satellite retrievals (Figure 7). The abrupt increasing $R_s$ trends of the observed $R_s$ from 1990 to 1994 are shown in zone V (southern China) and zone IX (Qinghai-Tibet Plateau), which might be caused by the

instrument replacements in this period [10]. Compared with other data, a high increasing trend after 2000 from the directly observed $R_s$ is shown in zones II, VI and VII. The SunDu-derived $R_s$ generally showed consistent $R_s$ variations with CERES in all regions. In zone II, Li et al. (2018) [81] reports a sharply increasing $R_s$ trend in the North China Plain due to controlling air pollution and reducing aerosol loading, which is consistent with our results. A slightly inconsistent $R_s$ variation in GEWEX is seen in zones I and V. It is probably because the high aerosols loading caused by factories and commercial activities in these regions, while GEWEX only use climatology aerosol data. Similar reasons might also result in an inconsistent $R_s$ variation in southern China from HXG, including zones III, IV, V and VI. By merging SunDu-derived $R_s$ data, HGWR can produce a consistent $R_s$ variation with the SunDu-derived $R_s$ and CERES. CMSAF have large different Rs variation compared with other datasets, including zones IV, VII, VIII and IX. This might because that large amount of invalid value in CMSAF especially before 2000.

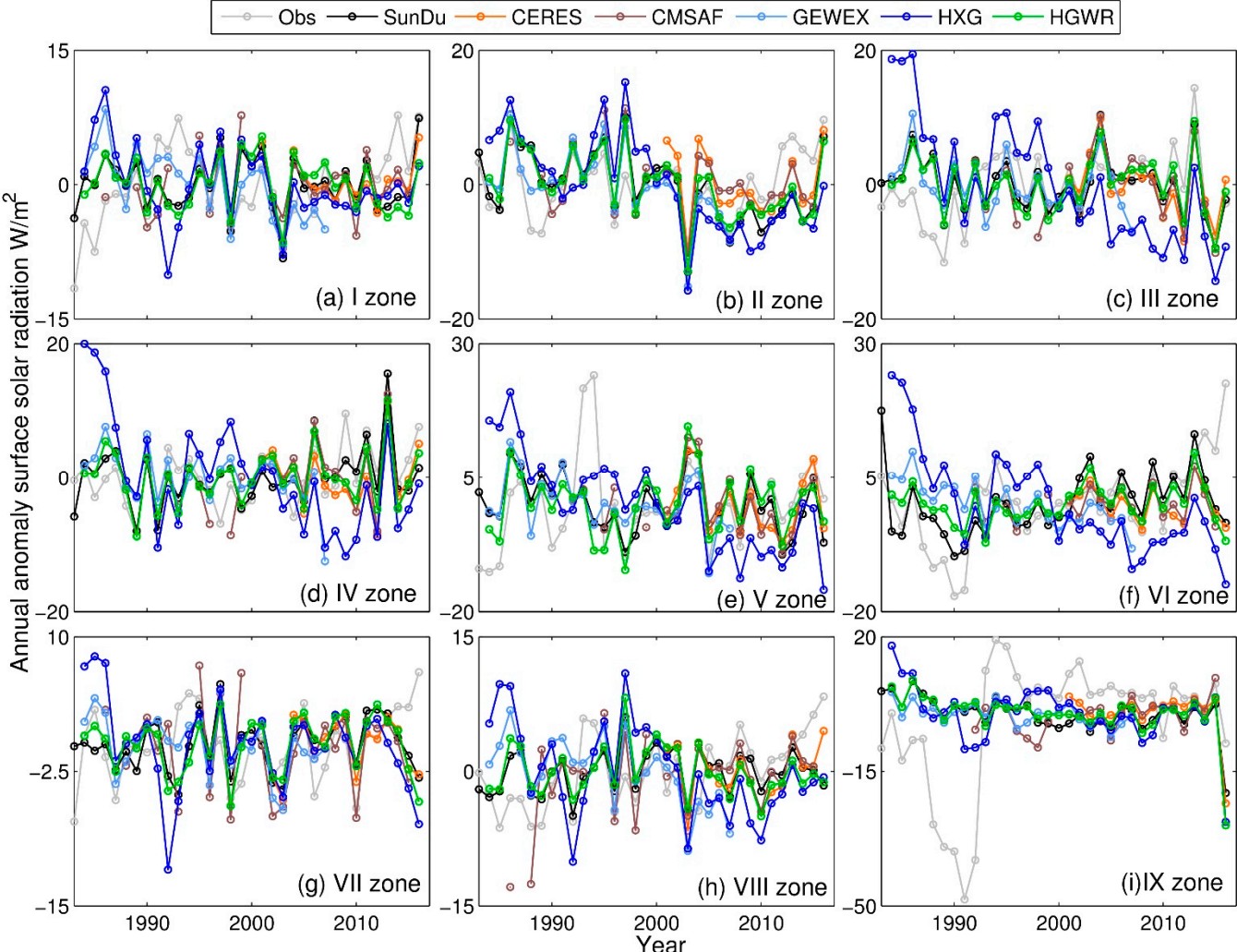

**Figure 7.** The regional mean of the annual anomaly of the $R_s$ for different subregions. Nine subregions (I to IX) over China are shown in Figure 1. The direct observed $R_s$ data, SunDu-derived $R_s$ and CERES EBAF are shown as the light grey lines, the black lines and the orange lines. The brown lines, the light blue lines and the blue lines represent the $R_s$ variation derived from the CMSAF CLARA A2 (CMSAF), GEWEX-SRB (GEWEX) and ISCCP-HXG derived $R_s$ (HXG), respectively. The ISCCP-HXG derived $R_s$ merged with SunDu derived $R_s$ (HGWR) is shown as the green lines.

## 4. Discussion

The results comparison indicates that the $R_s$ retrievals from HXG and HGWR show better performances than CMSAF and GEWEX. The global validation of hourly $R_s$ from HXG reported by (Tang et al. 2019) [57] suggests that HXG shows a better performance than ISCCP-FD. A comparison of GEWEX with direct observations over China suggests that GEWEX has large biases ranging from 8 to 9 W/m² [82]. Biases in cloud optical thickness have been shown to lead to large relative errors in the radiance fluxes over the Arctic when derived from GEWEX and CMSAF compared with CERES [83]. The validation results of HGWR illustrate that the impact of inhomogeneous input data can be reduced by merging with ground-based observations.

The GWR has been demonstrated to be quite flexible in many previous study areas, including geography [78,84], economics [85], meteorology [79,84,86] and epidemiology [87]. The results comparison from previous studies also suggests that GWR has a better performance than other methods, including the linear regression method and spline interpolation [79,88].

Previous studies have shown that $R_s$ over China has a decreasing trend from 1961 until approximately 1990 and levels off in approximately 1990 [42,89–91], which is consistent with our results. Except for the instrument replacement from 1990 to 1993 [10–12], the consistent trend from the direct observations and SunDu-derived $R_s$ indicates that the merged $R_s$ data by combined HXG and SunDu-derived $R_s$ can produce a reasonable long-term $R_s$ variation in China. Our previous study also demonstrates that the impact of aerosols can be included in SunDu derived $R_s$; the disadvantages of using constant aerosol value in $R_s$ satellite retrievals can be overcome by merging SunDu derived $R_s$ [59], and the product of this study is merged high resolution surface solar radiation data over 1983 to 2017 while the time period of products of [59] is from 2000 to 2017 based on MODIS cloud retrievals. We also compare the merged products with the MODIS based merged products of [59], all these datasets show generally similar $R_s$ trend from 2000 to 2016 (Figure S6). Slight differences exist which are closed with the relationship of input satellite data. We notice that the original MODIS cloud data are more spatially consistent, and the original ISCCP-HXG have more mosaic patterns. Most of these differences are eliminated by merged with SunDu derived $R_s$. The advantages of the MODIS based merged products is that the cloud and AOD are derived from MODIS retrievals. The advantage of the merged dataset from this study is the long-term record from 1983 to 2017.

The reduced performances of GEWEX and HXG might have been attributed to the inhomogeneity of cloud and aerosol input data in the radiation transfer model. Hayasaka (2016) [91] points out that discontinuities exist in the long-term variation in the $R_s$ from GEWEX-SRB due to the transition from one reference satellite to the next satellite. Wang et al. (2015) [10] also noticed that different amounts of data from polar orbits and geostationary satellites and their different capabilities for detecting low-level clouds introduced inhomogeneity into the ISCCP-derived $R_s$ data. For aerosols, the recent comparison work with CMSAF on the Iberian Peninsula for the period of 1985 to 2015 also suggests that the reason for the disagreement in trends between the satellite and reference ground-based SIS data over the Iberian Peninsula may be related to aerosols. Tang et al. (2019) [57] also noted that care should be taken when using HXG-derived $R_s$ for trend analysis due to the use of climatological aerosol and albedo data.

Uncertainties might be introduced when comparing ground observations with satellite data. However, Huang et al. (2016) [92] show that the sampling error for a $5 \times 5 \text{ km}^2$ area ranged from 1.4 to 8.1% on monthly to "instantaneous" timescales, respectively. The direct comparisons between the $R_s$ from the Geostationary Operational Environmental Satellite (GOES) and that measured at the Atmospheric Radiation Measurement (ARM) Southern Great Plains (SGP) site show that the sampling error decreases and levels off to a relatively small and stable value when averaged over periods greater than 5 days [93]. Hakuba et al. (2013a) [94] found that either a site-specific correction factor or increasing the number of sites in a grid box would reduce the sampling bias. These findings indicate

that the sampling error would be small on a monthly timescale. We notice that SunDu data are not evenly distributed, especially in the northwestern region. The accuracy of applying spatial interpolation methods will decrease significantly when ground stations are sparse and non-uniform [95]. GWR can overcome this deficiency of spatial interpolation by taking account of spatially heterogeneous relationships between the dependent variable and regressors. The results also demonstrate that our GWR based fusion methods have better performance in the northwestern region (Figure S1), However, the uncertainties cannot be ruled out entirely, which need more ground-based study in the northwestern region of China in the future.

## 5. Conclusions

We compared four long-term AVHRR $R_s$ products (GEWEX, CMSAF, HXG and HGWR) by using direct $R_s$ observations, SunDu-derived $R_s$ and CERES EBAF data as references. The spatial distribution of $R_s$ and long-term trends over China are evaluated from 1983 to 2016. The results show that the four $R_s$ products produce similar multi-year mean $R_s$ values, but there are large differences in simulating the $R_s$ variability and long-term trends.

Among these four $R_s$ products, HXG produced the longest $R_s$ records from 1983 to 2016 and few missing data. However, the long-term variability in $R_s$ from HXG is slightly inconsistent with the observations. Comparatively, HGWR shows the best performance with a high R2 and lower RMSE and has a consistent long-term $R_s$ trend and variability. The results suggest that further improvement in long-term trend $R_s$ simulations is still needed. The merged $R_s$ product (HGWR) can contribute to hydrology and ecology applications and research related to the simulation of land surface processes.

**Supplementary Materials:** The following are available online at https://www.mdpi.com/2072 -4292/13/4/602/s1. Figure S1: The regional validations of monthly surface solar radiation ($R_s$). Table S1: Details of the data products used in this comparison, Statistical summary of GWR parameter optimization. Table S3: Statistical comparison of surface solar radiation $R_s$.

**Author Contributions:** All of the authors collected data, designed the study, analyzed the data, and prepared the manuscript. All of the authors contributed to the main research ideas and manuscript organization. All of the authors thoroughly reviewed and edited this paper. All authors have read and agreed to the published version of the manuscript.

**Funding:** This study was funded by the Fundamental Research Funds for the Central Universities (#BLX201907), the National Key Research & Development Program of China (2017YFA06036001), the National Natural Science Foundation of China (41525018 and 41930970) and the State Key Laboratory of Earth Surface Processes and Resource Ecology (U2020-KF-02).

**Institutional Review Board Statement:** Not applicable.

**Data Availability Statement:** The merged high-resolution $R_s$ product from 1983 to 2017 over China, which can be downloaded at https://data.tpdc.ac.cn/zh-hans/data/a82849b0-9af5-457d-8968-447 1dd845f2e/.

**Acknowledgments:** This study was funded by the Fundamental Research Funds for the Central Universities (#BLX201907), the National Key Research & Development Program of China (2017YFA06036001), the National Natural Science Foundation of China (41525018 and 41930970) and the State Key Laboratory of Earth Surface Processes and Resource Ecology (U2020-KF-02). We would like to thank Yuna Mao, Jizeng Du, Runze Li, Qian Ma, Guocan Wu and Chunlue Zhou for their insightful comments. The CERES SYN data can be downloaded from https://ceres.larc.nasa.gov/data/.

**Conflicts of Interest:** The authors declare no conflict of interest.

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
