# Peer review of "Merging High-Resolution Satellite Surface Radiation Data with Meteorological Sunshine Duration Observations over China from 1983 to 2017"

_remotesensing, doi:10.3390/rs13040602_

Round 1

Reviewer 1 Report

In this work, a methodology is proposed to obtain an improved surface solar radiation product by using solar radiation data estimated from surface sunshine duration measurements as reference data. The study was based on the geographically weighted regression method, which was used to merge ISCCP-HXG-derived surface solar radiation with ground-based measurements over China. The work is well suited to the readership of RS. However, there are several issues with the procedure developed in the paper that prevent its publication in the current form.

The main issues with the work concern the terminology used in the title and throughout the manuscript: homogenizing. Homogenization implies something different than what is presented in the manuscript: the removal of unrealistic changes and jumps in the time-series mostly due to the change in instrumentation [1]. Usually, the break-points detected by the homogenization methods are carefully analysed together with metadata before removal. As the authors mentioned at line P5L184, in this work a data fusion was performed, more exactly, the ground-based data were spatially interpolated using as auxiliary information the ISCCP-HXG-derived solar radiation. As a consequence of that, the homogenization term must be changed throughout the manuscript and replaced with gridding, data fusion, merge etc.

Other issues:

P2L60.Please argument better this statement, by adding more relevant information from the literature. Satellite data are currently incorporated into numerical weather models, hence they could provide also accurate spatial estimates of the surface solar radiation.

P2L63. Provide actual references.

P5L204. Define collocated; are the Rs values similar for collocated stations? Also, add the number of sites whose data were used for validation.

P7L260.It would be interesting to see if the situation will remain the same if the scatterplots are realised at a seasonal scale, at least for the data available for a longer period of time (GEWEX, CMSAF, ISCCP). So, the seasonal scatterplots realised from monthly data should be added as ESM, with the corresponding accuracy indicators.

P7L268. Something is wrong with the reference to the table, perhaps the authors are referring to table S3

P8L275, Add units of measure to the legend. The same correction must be done to Figure 2.

P18L636 Update the Table S1 with CERES EBAF details.

  1. https://library.wmo.int/doc_num.php?explnum_id=9252

Author Response

Reviewer #1

Major comments:

  • In this work, a methodology is proposed to obtain an improved surface solar radiation product by using solar radiation data estimated from surface sunshine duration measurements as reference data. The study was based on the geographically weighted regression method, which was used to merge ISCCP-HXG-derived surface solar radiation with ground-based measurements over China. The work is well suited to the readership of RS. However, there are several issues with the procedure developed in the paper that prevent its publication in the current form. The main issues with the work concern the terminology used in the title and throughout the manuscript: homogenizing. Homogenization implies something different than what is presented in the manuscript: the removal of unrealistic changes and jumps in the time-series mostly due to the change in instrumentation [1]. Usually, the break-points detected by the homogenization methods are carefully analysed together with metadata before removal. As the authors mentioned at line P5L184, in this work a data fusion was performed, more exactly, the ground-based data were spatially interpolated using as auxiliary information the ISCCP-HXG-derived solar radiation. As a consequence of that, the homogenization term must be changed throughout the manuscript and replaced with gridding, data fusion, merge etc.

Reply: The authors would like to thank the reviewer #1 for his detailed and helpful comments. We have revised the manuscript followed the reviewer comments. We have realized that the definition of homogenization is removal of unrealistic changes and jumps in the time-series mostly due to the change in instrumentation and might not would be appropriate to describe the work of our study. According to the suggestion, we have replaced the homogenization term throughout the manuscript with other words, such as gridding, data fusion, merge. The title has been changed to “Merging high-resolution satellite surface radiation data with meteorological sunshine duration observations over China from 1983 to 2017”. The point by point responses of detailed comments are provided below.

Minor comments:

  • Please argument better this statement, by adding more relevant information from the literature. Satellite data are currently incorporated into numerical weather models, hence they could provide also accurate spatial estimates of the surface solar radiation.

Reply: We have added more relevant information for improvement of assimilating Satellite data into numerical weather models. (Lines 60-63):

“Improvement have also been shown in Rs data from DISORT radiation transfer model of the China Meteorological Administration Land Data Assimilation System (CLDAS) by assimilating satellite Rs retrievals from FY2E and ISCCP cloud data[28].”

  • Provide actual references.

Reply: The references have already been cited in [29] (Line 65).

  • Define collocated; are the Rs values similar for collocated stations? Also, add the number of sites whose data were used for validation.

Reply: We have revised the description of Rs validations sites and added the number of sites for validation. (Lines 209-211):

“All the 121 direct Rs observation sites, where both direct observed Rs data and SunDu-derived Rs data are recorded, are used as validation sites, and the remaining 2261 sites were merged with HXG by GWR (Figure 1).”

  • It would be interesting to see if the situation will remain the same if the scatterplots are realised at a seasonal scale, at least for the data available for a longer period of time (GEWEX, CMSAF, ISCCP). So, the seasonal scatterplots realised from monthly data should be added as ESM, with the corresponding accuracy indicators.

Reply: Thanks your suggestions. Figure 2 is the scatterplots of monthly Rs with corresponding statistics, indicating the performances of these satellite Rs retrievals (GEWEX, CMSAF, ISCCP) at seasonal scale.

  • Something is wrong with the reference to the table, perhaps the authors are referring to table S3

Reply: Thanks your suggestions. We have corrected the mistakes. (Lines 234-236).

“When comparing with SunDu-derived Rs, GEWEX has the largest MAB and RMSE (17.46 W/m2 and 22.17 W/m2, respectively) followed by CMSAF and HXG (table S3).”

  • P8L275, Add units of measure to the legend. The same correction must be done to Figure 2.

Reply: Corrected as suggestions (Lines 262 and 277). The units of the figure 2 and figure 3 have been added.

Figure 2. Comparison of monthly surface solar radiation (Rs) from the CMSAF CLARA-A2 (CMSAF), GEWEX-SRB (GEWEX), ISCCP-HXG-based Rs data (HXG) and the merged product (HGWR) by using different validation data from 2000 to 2007. Subplots (a, d, g, j) represent validation results using direct observations, while (b, e, h, k) represent SunDu-derived Rs data (SunDu Rs) and subplots (c, f, i, l) represent CERES EBAF data (CERES).

Figure 3. Spatial distribution of multiyear mean Rs from 2000 to 2007. The first line (a, b) shows the observed multiyear mean Rs from SunDu and CERES EBAF (CERES); the multiyear mean Rs derived from the CMSAF CLARA-A2 (CMSAF) and GEWEX-SRB (GEWEX) are shown in the second line (c, d), respectively. The third line (e, f) shows the observed multiyear mean monthly Rs from ISCCP-HXG (HXG) and ISCCP-HXG merged with SunDu-derived Rs (HGWR).

  • P18L636 Update the Table S1 with CERES EBAF details.

Reply: Corrected as suggestions in Table S1.

Reviewer 2 Report

This manuscript presents a comparison study of three satellite surface solar radiation (Rs) products based on AVHRR cloud products with direct observations of Rs and sunshine duration (SunDu)-derived Rs. The authors also proposed a geographically weighted regression (GWR) fusion method to merge the International Satellite Cloud Climatology Project (ISCCP) HXG cloud products Rs with SunDu-derived Rs. When I read through the manuscript, I found this manuscript is highly similar to a manuscript (“Merging ground-based sunshine duration with satellite cloud and aerosol data to produce high resolution long-term surface solar radiation over China” doi:10.5194/essd-2020-231) submitted to another journal by the same authors. In that manuscript, the authors also applied GWR to fuse the cloud products with the SunDu-drived Rs. The figures are also quite similar to that manuscript. Is this manuscript a resubmission to Remote Sensing? If not, the authors need to clarify the difference and what additional contributions this manuscript makes to current literature. The connections with the authors’ previous studies should be mentioned in the introduction section and discussed in the discussion section. Without the clarification, the manuscript should not be considered for publication. The results section needs to be improved. The current version is just repeating numbers, there is no in-depth analysis.

The author uses GWR to fuse the ISCCP-HXG-derived Rs with ground-based SunDu-derived Rs. The section 2.3 is not quite clear on how GWR was exactly used. Is the dependent variable the ground-based SunDu Rs? and the independent variable(s) is the ISCCP-HXG Rs? The obtained weights are just the regression coefficients between the SunDu Rs and the ISCCP-HXG Rs of the neighborhood? So the fusion results are calculated based on the regression weights and the ISCCP-HXG Rs? It is not clear in the manuscript, the authors need to provide more detailed descriptions on how the GWR was used? How was GWR implemented? ArcGIS?

Figure 1 shows that the SunDu data are not evenly distributed over China, especially that the data are quite sparse in the northwestern region. How does this uneven distribution affect the GWR results and the fusion results? This should be discussed as the data fusion uncertainty.

I found that the results section was just literally repeating the numbers of the table, and superficially describing the figures. It’s hard for readers to follow the main results in those repeating numbers that were already shown on the table. The authors need to improve the results section by summarizing some major statements on what they found, and interpreting their findings instead of repeating numbers.

Figure 5: the spatial distribution pattern of trends is quite different between these products. However, the authors didn’t analyze the difference at all. L279-L287 needs to be expanded to analyze figure 5 in-depth, like which regions have the largest difference? What are the potential reasons for explaining the difference?

Figure 7: again the authors need to analyze the discrepancies between different datasets for different regions, and the potential reasons.

L240: are based on…

L245: indicating -> indicates

Author Response

Reviewer #2

Major comments:

  • This manuscript presents a comparison study of three satellite surface solar radiation (Rs) products based on AVHRR cloud products with direct observations of Rs and sunshine duration (SunDu)-derived Rs. The authors also proposed a geographically weighted regression (GWR) fusion method to merge the International Satellite Cloud Climatology Project (ISCCP) HXG cloud products Rs with SunDu-derived Rs. When I read through the manuscript, I found this manuscript is highly similar to a manuscript (“Merging ground-based sunshine duration with satellite cloud and aerosol data to produce high resolution long-term surface solar radiation over China” doi:10.5194/essd-2020-231) submitted to another journal by the same authors. In that manuscript, the authors also applied GWR to fuse the cloud products with the SunDu-drived Rs. The figures are also quite similar to that manuscript. Is this manuscript a resubmission to Remote Sensing? If not, the authors need to clarify the difference and what additional contributions this manuscript makes to current literature. The connections with the authors’ previous studies should be mentioned in the introduction section and discussed in the discussion section. Without the clarification, the manuscript should not be considered for publication. The results section needs to be improved. The current version is just repeating numbers, there is no in-depth analysis.

Reply: Thanks for your comments and suggestions. This manuscript is not a resubmission. This manuscript is different from the ESSD manuscript (doi:10.5194/essd-2020-231). The work of this manuscript is to directly merge SunDu-derived Rs with satellite Rs retrievals and compare with current Rs products. While work of the ESSD manuscript is to merge SunDu-derived Rs with satellite cloud and aerosols retrievals and analyses the different Rs impact factors for merging. The product of this study is merged high resolution surface solar radiation data over 1983 to 2017 while the time period of products of the ESSD paper is much shorter, from 2000 to 2017. We have modified the introduction section and discussed in the discussion section to clarify the connection with our previous studies (Lines 114-119 and lines 396-400).

“Our previous studies have shown reanalyses have substantial biases due to imperfect parameterization of cloud and aerosols [26,52,59]. Considering the advantages of SunDu-derived Rs and reliable cloud distribution from satellite retrievals, we recently combine ground-based sunshine duration with satellite cloud and aerosol data to produce high resolution long-term Rs [60]. Another approach is to directly merge current high resolution Rs product with SunDu-derived Rs.”

“Our previous study also demonstrate that the impact of aerosols can be included in SunDu derived Rs and the disadvantages of using constant aerosol value in Rs satellite retrievals can be overcome by merging SunDu derived Rs and the product of this study is merged high resolution surface solar radiation data over 1983 to 2017 while the time period of products of [60] is from 2000 to 2017 based on MODIS cloud retrievals.”

Below are our point by point responses to your comments.

Minor comments:

  • The author uses GWR to fuse the ISCCP-HXG-derived Rs with ground-based SunDu-derived Rs. The section 2.3 is not quite clear on how GWR was exactly used. Is the dependent variable the ground-based SunDu Rs? and the independent variable(s) is the ISCCP-HXG Rs? The obtained weights are just the regression coefficients between the SunDu Rs and the ISCCP-HXG Rs of the neighborhood? So the fusion results are calculated based on the regression weights and the ISCCP-HXG Rs? It is not clear in the manuscript, the authors need to provide more detailed descriptions on how the GWR was used? How was GWR implemented? ArcGIS?

Reply: We have refined detailed descriptions of processes and implementation of GWR in our merging method (Lines 197-205).

“where wij is the weighting function for SunDu derived Rs observation site j that refers to location i; dij denotes the Euclidian distance between j and i; and b is the size of the neighborhood and the maximum distance away from regression location i, known as “bandwidth”, which is determined by the number of nearest neighbor points (NNPs). The NNPs are set to 30 following our parameterization experiment (Table S2, Feng and Wang, 2020 under review). The regression coefficient are then determined by equation (3) and (4) using the ground-based SunDu Rs as the dependent variable and ISCCP-HXG Rs at corresponding observation site as the independent variable. The final merged Rs can be obtain by the regression coefficient and ISCCP-HXG Rs for the whole regions by equation (1), which can be applied in ArcGIS or R.”

  • Figure 1 shows that the SunDu data are not evenly distributed over China, especially that the data are quite sparse in the northwestern region. How does this uneven distribution affect the GWR results and the fusion results? This should be discussed as the data fusion uncertainty.

Reply: Thanks for your comments. We have added the uncertainties of impact of uneven distribution sites in the discussion section (Lines 422-429).

“We notice that SunDu data are not evenly distributed, especially in the northwestern region. The accuracy of applying spatial interpolation methods will decrease significantly when ground stations is sparse and non-uniform [95]. GWR can overcome this deficiency of spatial interpolation by taking account of spatially heterogeneous relationships between the dependent variable and regressors. The results also demonstrate that our GWR based fusion methods have better performance in the northwestern region (Figure S1), However, the uncertainties can't be ruled out entirely, which need more ground-based study in the northwestern region of China in the future.”

  • I found that the results section was just literally repeating the numbers of the table, and superficially describing the figures. It’s hard for readers to follow the main results in those repeating numbers that were already shown on the table. The authors need to improve the results section by summarizing some major statements on what they found, and interpreting their findings instead of repeating numbers.

Reply: Thanks for your comments. We have revised the results section to make the major statements more clear and add the interpretation of our findings. The major changes are shown below and the changes of analysis for figure 4, 5 and 7 are shown in following comments.

(Lines 230-238):

“All these four AVHRR-based Rs products showed high R2 values ranging from 0.89 to 0.97 (Figure 2). Generally, HGWR show best performances followed by HXG, GEWEX and CMSAF. Specifically, CMSAF has the largest MAB and RMSE (18.96 W/m2 and 24.08 W/m2, respectively) followed by GEWEX and HXG, compared with direct observations. When comparing with SunDu-derived Rs, GEWEX has the largest MAB and RMSE (17.46 W/m2 and 22.17 W/m2, respectively) followed by CMSAF and HXG. Similar results can also be seen when validating against the SunDu-derived Rs. The good performances of HXG might be attributed to the improvements of high resolution of cloud from latest ISCCP-H series product and auxiliary data from ERA5.”

(Lines 252-254):

“These results suggest that without the SunDu derived Rs constraint, these AVHRR based satellite Rs retrievals still have uncertainties in simulating aerosols.”

(Lines 257-260):

“The areas with significant improvement of HXG are mainly distributed in east part of China, which might be the improvement in simulating the impact of aerosols loading by merging SunDu derived Rs.

(Lines 273-274):

“which might be attributed to the high spatial resolution of cloud input data from ISCCP-HXG (Table S3).”

  • Figure 5: the spatial distribution pattern of trends is quite different between these products. However, the authors didn’t analyze the difference at all. L279-L287 needs to be expanded to analyze figure 5 in-depth, like which regions have the largest difference? What are the potential reasons for explaining the difference?

Reply: we have added more analysis of the spatial distribution pattern of trends is quite different between these products and the potential reasons for the differences for different regions (Lines 291-300).

“The difference spatial distribution of Rs trend might be attributed to the input of cloud and aerosols data. Both CERES EBAF and CMSAF use CALIPSO-CALIOP cloud information to adjust the cloud input data. Both GEWEX and HXG use climatology of aerosol data [22]. In southern coastal regions of China, all Rs products show decreasing trend, which might be the dominated impacts of clouds. However, in the southern inland regions of China, the Rs trend from these Rs products are quite different, which might be the different ability of simulating the aerosols loading in these regions. In west part of China, largest differences of Rs trend from these Rs products show in Qinghai-Tibet Plateau and Tianshan Mountains, which might be attributed to impacts of the complex terrain. Similar results also show in Daxinganling Mountain in Northeast China.”

  • Figure 7: again the authors need to analyze the discrepancies between different datasets for different regions, and the potential reasons.

Reply: we have added more analysis the discrepancies between different datasets for different regions, and the potential reasons for the discrepancies as below:

(Lines 350-352):

The abrupt increasing Rs trends of the observed Rs from 1990 to 1994 are shown in zone V (southern China) and zone IX (Qinghai-Tibet Plateau), which might be caused by the instrument replacements in this period [10].

(Lines 357-360):

It's probably because the high aerosols loading caused by factories and commercial activities in these regions, while GEWEX only use climatology aerosol data. Similar reasons might also results in an inconsistent Rs variation in southern China from HXG, including zones III, IV, V and VI.

(Lines 362-364):

CMSAF have large different Rs variation compared with other datasets, including zones IV, VII, VIII and IX. This might because that large amount of invalid value in CMSAF especially before 2000.

  • L240: are based on…

Reply: Corrected as suggestions (Lines 242).

  • L245: indicating -> indicates

Reply: Corrected as suggestions (Lines 246).

Round 2

Reviewer 1 Report

The authors generally did a good job in answering and accounting for my concerns of the originally submitted manuscript. Nevertheless, I think that one of the reviewer's major concerns were not appropriately addressed:

I asked the authors to analyse the data using the scatterplots (figure 2) at the seasonal scale to verify if the results are consistent also on subsets of the data; although the authors answered that the scatterplots are realised using data at a seasonal scale, they provided in the revised version of the manuscript the same graphs as in the originally submitted of the manuscript.

Author Response

The authors would like to thank the reviewer #1 for his detailed comments. We have realized that we may not catch your meaning of data on seasonal scale. We have revised the manuscript followed your comments. We have drawn the scatterplots for different seasonal and put these figures in the supplement file. “Similar results can also been seen in different seasonal (Figure S2-S5).” (Line 241). The figures are shown as below: 

Reviewer 2 Report

In this revised version, the authors have improved the writing of results section. However, I still have concerns regarding the similarity of this manuscript with another manuscript that is under review “Merging ground-based sunshine duration with satellite cloud and aerosol data to produce high resolution long-term surface solar radiation over China”. Even though these two manuscripts used different strategies of fusing data, one is using cloud and aerosol data as covariables to fuse data, another one (this manuscript) is directly merging SunDu-derived Rs with satellite Rs retrievals. From my point of view, these two manuscripts should be merged into one manuscript given the same study objective and similar data products and also these two manuscripts were almost submitted to different journals at the same time. If this manuscript has a great improvement over another one, the authors need to compare these fused datasets over the overlapped time periods, and give discussion regarding the difference and improvement of these two different fusion strategies. I understand that it is always challenging to compare with other people’s study when proposing a new method, but the authors have both fused datasets, and should give a comparison in this study. If there are discrepancies between these two datasets, the authors should give explanations and discussions since the two datasets were generated by the same research group. If the authors think the temporal coverage is the main difference between these two datasets, the authors should compare the datasets in the overlapped period and apply a bias correction to merge the two datasets.

Author Response

Major comments:

  • In this revised version, the authors have improved the writing of results section. However, I still have concerns regarding the similarity of this manuscript with another manuscript that is under review “Merging ground-based sunshine duration with satellite cloud and aerosol data to produce high resolution long-term surface solar radiation over China”. Even though these two manuscripts used different strategies of fusing data, one is using cloud and aerosol data as covariables to fuse data, another one (this manuscript) is directly merging SunDu-derived Rswith satellite Rs From my point of view, these two manuscripts should be merged into one manuscript given the same study objective and similar data products and also these two manuscripts were almost submitted to different journals at the same time. If this manuscript has a great improvement over another one, the authors need to compare these fused datasets over the overlapped time periods, and give discussion regarding the difference and improvement of these two different fusion strategies. I understand that it is always challenging to compare with other people’s study when proposing a new method, but the authors have both fused datasets, and should give a comparison in this study. If there are discrepancies between these two datasets, the authors should give explanations and discussions since the two datasets were generated by the same research group. If the authors think the temporal coverage is the main difference between these two datasets, the authors should compare the datasets in the overlapped period and apply a bias correction to merge the two datasets.

Reply: Thanks for your comments. We compare the datasets in the overlapped period from 2000 to 2016. The spatial distributions of trends for Rs from 2000 to 2016 derived from these datasets are shown in figure S6. The GWR-MODIS-CF-AOD and GWR-MODIS-CF are the datasets from another manuscript. GWR-MODIS-CF-AOD is the datasets that merged MODIS cloud and AOD with SunDu derived Rs, while GWR-MODIS-CF is the dataset that merged only MODIS cloud with SunDu derived Rs. The GWR-ISCCP-HXG is the merged dataset from this study. Generally all these datasets shown consistent spatial pattern of Rs trend, especially in eastern China. Slightly differences exist which are closed with the relationship of input satellite data. We notice that the original MODIS cloud data are more spatially consistent and the original ISCCP-HXG have more mosaic patterns. Most of these differences are eliminated by merged with SunDu derived Rs. We also add this part in the revised manuscript (Lines 401-407):

“We also compare the merged products with the MODIS based merged products of [60], all these datasets show generally similar Rs trend from 2000 to 2016 (Figure S6). Slightly differences exist which are closed with the relationship of input satellite data. We notice that the original MODIS cloud data are more spatially consistent and the original ISCCP-HXG have more mosaic patterns. Most of these differences are eliminated by merged with SunDu derived Rs. The advantages of the MODIS based merged products is that the cloud and AOD are derived from MODIS retrievals. The advantages of merged dataset from this study is the long term record from 1983 to 2017.”
